# Volatilomics: An Emerging and Promising Avenue for the Detection of Potential Prostate Cancer Biomarkers

**DOI:** 10.3390/cancers14163982

**Published:** 2022-08-17

**Authors:** Cristina V. Berenguer, Ferdinando Pereira, Jorge A. M. Pereira, José S. Câmara

**Affiliations:** 1CQM—Centro de Química da Madeira, NPRG, Universidade da Madeira, Campus da Penteada, 9020-105 Funchal, Portugal; 2SESARAM—Serviço de Saúde da Região Autónoma da Madeira, EPERAM, Hospital Dr. Nélio Mendonça, Avenida Luís de Camões 6180, 9000-177 Funchal, Portugal; 3Departamento de Química, Faculdade de Ciências Exatas e Engenharia, Universidade da Madeira, Campus da Penteada, 9020-105 Funchal, Portugal

**Keywords:** prostate cancer, diagnosis, volatilomics, tumor biomarkers

## Abstract

**Simple Summary:**

The lack of highly specific and sensitive biomarkers for the early detection of prostate cancer (PCa) is a major barrier to its management. Volatilomics emerged as a non-invasive, simple, inexpensive, and easy-to-use approach for cancer screening, characterization of disease progression, and follow-up of the treatment’s success. We provide a brief overview of the potential of volatile organic metabolites (VOMs) for the establishment of PCa biomarkers from non-invasive matrices. Endogenous VOMs have been investigated as potential biomarkers since changes in these VOMs can be characteristic of specific disease processes. Recent studies have shown that the conjugation of the prostate-specific antigen (PSA) screening with other methodologies, such as risk calculators, biomarkers, and imaging tests, can attenuate overdiagnosis and under-detection issues. This means that the combination of volatilomics with other methodologies could be extremely valuable for the differentiation of clinical phenotypes in a group of patients, providing more personalized treatments.

**Abstract:**

Despite the spectacular advances in molecular medicine, including genomics, proteomics, transcriptomics, lipidomics, and personalized medicine, supported by the discovery of the human genome, prostate cancer (PCa) remains the most frequent malignant tumor and a leading cause of oncological death in men. New methods for prognostic, diagnostic, and therapy evaluation are mainly based on the combination of imaging techniques with other methodologies, such as gene or protein profiling, aimed at improving PCa management and surveillance. However, the lack of highly specific and sensitive biomarkers for its early detection is a major hurdle to this goal. Apart from classical biomarkers, the study of endogenous volatile organic metabolites (VOMs) biosynthesized by different metabolic pathways and found in several biofluids is emerging as an innovative, efficient, accessible, and non-invasive approach to establish the volatilomic biosignature of PCa patients, unravelling potential biomarkers. This review provides a brief overview of the challenges of PCa screening methods and emergent biomarkers. We also focus on the potential of volatilomics for the establishment of PCa biomarkers from non-invasive matrices.

## 1. Introduction

Prostate cancer (PCa) is the second most frequent malignant tumor, the fifth leading cause of cancer death among men worldwide (the leading cause of cancer death among men in 46 countries), and the most frequently diagnosed cancer in 105 of 185 of the world countries [1]. In 2020, almost 1.4 million new cases and about 0.4 million deaths were estimated (GLOBOCAN data) [2]. PCa is very heterogeneous in terms of grade and genetics, displaying complex biological, hormonal, and molecular features [2]. This cancer has different phenotypes, ranging from indolent asymptomatic, a non-life-threatening form, to metastatic, very aggressive, rapidly progressive, and lethal forms [3,4]. Unlike diseases such as breast and colon cancer, no major predisposition genes for PCa have been detected. Instead, multiple chromosomal loci of susceptibility genes have been identified, and most of the genomic regions remain poorly studied, which explains this cancer’s heterogeneity [5]. Furthermore, epigenetic factors play an important role in its clinical phenotypes [6].

PCa and subsequent treatments have a high impact on the functional and psychological status of patients, significantly affecting their quality of life [7]. The current diagnostic methods are based on the measurement of prostate-specific antigen (PSA) blood levels, transrectal ultrasound, digital rectal examinations (DRE), and prostate biopsies [8] (Figure 1). However, these methods are invasive, expensive, and unpleasant to patients, with consequent risks of unnecessary complications, and can lead to both false-positive and false-negative results [9]. The PSA test has limited sensitivity (20.5%) [10], accuracy (62–75%) [11], and specificity (51–91%) [12]. Its low selectivity to detect PCa often leads to the overdiagnosis and overtreatment of relatively indolent tumors with low potential for morbidity or death if left untreated [13].

The advances in OMICs science have contributed to the discovery of new biomarkers for PCa detection, management, and surveillance. Despite the great efforts and important discoveries, no biomarker has been able to replace PSA in clinical practice for PCa screening [10]. Hence, need is urgent to find highly specific diagnostic tools for non-invasive detection of PCa that are preferentially able to stratify patients by cancer aggressiveness and consequent choice of therapy, which will lead to personalized and targeted therapies. More recently, volatilomics emerged as a promising approach for the definition of cancer biomarkers, based on metabolites biosynthesized by different metabolic pathways, and found in readily accessible biofluids, such as saliva, exhaled breath, and urine. In this review, we provide a brief overview of the potential of endogenous volatile organic metabolites (VOMs) as an innovative and efficient approach to establishing a volatilomic biosignature and potentially define a panel of PCa biomarkers as a complementary tool to be used in clinical practice for its diagnostic and management.

## 2. Prostate Cancer Biomarkers

In recent years, advances in molecular medicine have contributed to the discovery of new potential biomarkers to aid in PCa screening and management. Common liquid biopsies biomarkers include extracellular vesicles (EVs), circulating tumor cells (CTCs) and DNA (ctDNA), and cell-free DNA (cfDNA) [14]. However, a few issues prevent the effective use of CTCs and EVs as biomarkers in liquid biopsies for diagnosing PCa, such as the need for specific guidelines for the biomarker’s isolation and detection. Moreover, the microfluidic devices used to develop liquid biopsies have not yet been fully validated and standardized [15]. Long noncoding RNAs (lncRNAs) have also emerged as a promising class of PCa biomarkers. Most lncRNAs associated with PCa are overexpressed in tumor tissues and cancer cells, contributing to tumor proliferation, invasion, and metastasis. In contrast, only a few lncRNAs are downregulated and may act as tumor suppressors, in addition to their potential activity as transcriptional regulators and oncogenes. All of these unique features make lncRNAs promising predictive biomarkers and therapeutic targets for the diagnosis, screening, prognosis, and progression of PCa. Nevertheless, the molecular mechanisms of action of lncRNAs are not very clear yet and it will be important to fully understand and investigate the roles and mechanisms of lncRNAs in prostate carcinogenesis [16]. Other molecular biomarkers for urine, serum, and tissue samples have been developed (Table 1) based on the combination of imaging techniques with other methodologies, such as gene or protein profiling, to enhance cancer detection, pre-biopsy decision, determination of cancer risk, and therapeutic management of PCa [17].

Abnormalities in these tests indicate the performance of a prostate biopsy. Moreover, risk calculators are combined with these tests to help determine the risk of cancer, thus reducing the number of unnecessary biopsies. The guidelines recommend using these tests in combination with the current PCa screening methods (Figure 2) [10].

Biomarkers such as the lncRNA PCA3 and TMPRSS2-ERG fusion gene have shown increased sensitivity and specificity (Table 1), potentially reducing PCa overdiagnosis. The prostate cancer gene 3 (PCA3) assay detects long non-coding RNA in urine samples. This test was approved by the Food and Drug Administration (FDA) in 2012; it calculates the ratio of PCA3 messenger RNAs (mRNAs) versus PSA mRNA in the first urine post-DRE and is approved for patients with a previously negative biopsy [7,10,32,33]. The Prostate Health Index (PHI) test is an algorithm approved by the FDA that includes total PSA, free PSA, and p2PSA isoform ([-2] proPSA). PHI calculates PCa probability and is recommended for men with PSA levels between 2 and 10 ng/mL and no abnormalities in their DRE. This blood test is also able to assess the likelihood of PCa progression during active surveillance, being used to monitor patients [34]. The Four-Kallikrein (4KScore) test is a diagnostic algorithm that combines four kallikreins in blood plasma, namely human kallikrein 2 (hK2), total PSA, free PSA, intact PSA, in addition to the patient’s clinical information (age, DRE results, and prostate biopsy history). This test assesses the probability of high-grade PCa and is recommended for patients undergoing initial and repeated biopsy. Moreover, the 4KScore also predicts the risk of occurrence and development of aggressive PCa [34]. The ExoDx Prostate IntelliScore (EPI) is a pre-biopsy RNA-based assay that uses the expression of PCA3, ERG, and SPDEF, isolated from urinary exosomes to predict the probability of high-grade PCa (Gleason score ≥ 7) on diagnostic biopsy. This is the only test that is not based on any other parameters related to PSA or a PSA derivative in the test algorithm to calculate the result, but clinicians can use it in conjugation with other clinical variables [30,34]. The SelectMDx and MyProstateScore (MPS) tests are based on the combination of multiple gene analyses. SelectMDx is a non-invasive test that measures mRNA transcripts from the genes HOXC6 and DLX1 in urine samples post-DRE and relates them to clinical risk factors such as age, family history, and PSA levels. This test is used to evaluate the presence of any PCa during biopsy and the risk of high-grade PCa. It also avoids unnecessary biopsies in the case of low-risk PCa [27]. Serum and urine biomarkers are used for consideration of initial biopsy, while tissue biomarkers are used to confirm test results. Tissue biomarkers tests have been developed to aid the clinical practice to decide what kind of therapy should be applied for different PCa diagnoses. ConfirmMDx is based on pronounced epigenetic changes that are indicative of the presence of cancer in the benign prostate tissue that is near the focus of PCa. This test determines the level of methylation of the promoter regions of the genes GSTP1, APC, and RASSF1 in benign prostate tissue, identifying high-grade PCa in patients with negative biopsies [27]. Prolaris is a prognostic test that measures tumor biology to improve the accuracy of risk stratification in men with localized PCa. This test combines the RNA expression levels of 31 genes involved in cell-cycle progression and 15 housekeeping genes to generate a Prolaris Score. Prolaris can be used to guide patient selection for active surveillance or definitive treatment [34,35,36]. The ProMark test is a protein-based prognostic assay of eight protein markers (DERL1, CUL2, SMAD4, PDSS2, HSPA9, FUS, phosphorylated S6, and YBOX1) that predicts the aggressiveness of cancer in patients with Gleason scores of 3 + 3 and 3 + 4. Moreover, ProMark predicts adverse pathology during radical prostatectomy and predicts if the tumor can be managed with or without aggressive treatment [27]. The Decipher is a genomic classifier prediction model for metastasis that measures the levels of RNA expression of 22 different genes on post-prostatectomy tissue samples. This test calculates the likelihood of clinical metastases within 5 years of prostatectomy in men with adverse pathological features. It could be a useful tool for diagnosis and local therapy planning for new PCa patients [31,34].

Despite the recent progress in the discovery of new biomarkers, gene mutations, and genomic signatures, some challenging obstacles must be overcome to develop effective biomarkers. These limitations include tumor heterogeneity, tumor–host interplay, complexity, multiplicity, and redundancy of tumor-cell signaling networks, involving genetic, epigenetic, and microenvironmental effects [7]. Additionally, the technologies associated with these approaches are often expensive, unavailable in many medical facilities, and time-consuming [10].

## 3. Contribution of the OMICs Science

The OMICs science comprises the dataset of genomics (DNA), transcriptomics (RNA), proteomics (proteins), and metabolomics (metabolites) and is intended to be used to discover cancer-specific biomarkers that are useful for its diagnosis and prognosis. In recent years, metabolomics emerged as a promising tool to offer novel insights into disease aetiology and etiological pathways [37]. Metabolomics is complementary to genomics, transcriptomics, and proteomics, as it represents the integration of genetic regulation, enzyme activity, and metabolic reactions [38]. This science studies the complex interaction of small molecules in biological systems providing comprehensive and detailed information on the phenotype and molecular physiology changes resultant from the interactions between environmental factors, genetics, and both exogenous and endogenous factors, such as age, diet, drugs, chronobiological variations, among others [7]. Metabolites represent the end-products of physiological processes, and the altered levels of certain metabolites can be measured by using metabolomics to establish a correlation with the disease status [39]. Neoplastic cells have a unique metabolic phenotype that is related to cancer development and progression, resulting in changes in the production, use, and levels of metabolites [37,39,40]. Therefore, metabolomics has become a powerful tool for the discovery of new cancer biomarkers and therapeutic monitoring through the analysis of biomarkers indicative of disease progression and therapeutic response [41,42].

Volatilomics is a subset of metabolomics based on the study of VOMs. VOMs are low-molecular-weight metabolites (<500 Da) with high volatility and a carbon-based chemical group. These metabolites are generated through the metabolism of cells, reflecting their biological activity, and can be released in the blood and excreted through exhaled breath, sweat, urine, or saliva [43]. Cancer cells can be distinguished from normal cells by alterations in normal metabolic rates, apoptotic pathways, and protein expression patterns [10,43,44]. Metabolic shifts and different responses of the immune system may consist of some of the earliest and most detectable changes in cancer which may become more pronounced as the disease develops. Since VOMs are produced and emitted through the metabolism of cells, cancer development and progression can lead to changes in the volatilomic profile which can be used to define a volatilomic biosignature for diagnostic purposes [10,39].

### 3.1. Volatilomics—An Emerging Yet Challenging Approach

Empirical data have confirmed the potential of VOMs analysis for cancer screening, characterization of disease progression, and follow-up of the treatment’s success, as well as for the discrimination between different types of cancer. The volatilomics approach is based on highly sensitive analytical techniques and does not require invasive procedures, since VOMs can be found in readily accessible biofluids.

Volatilomics studies can range from targeted analysis of one or a small number of metabolites associated with a specific biological pathway to the fingerprinting of a large subset of metabolites associated with a particular phenotype or stimulus [38,44,45]. Untargeted approaches are more appropriate to detect unexpected changes in the concentrations of specific metabolites [7], meaning that the use of a multi-biomarker panel provides a better evaluation of the cancer progression [10]. VOMs’ detection requires precise, reliable, and effective instrumentation (Figure 3) [46]. Mass spectrometry (MS) is the most used analytical platform for the identification of the volatilomic profile of biological matrices. MS requires an initial separation of metabolites by gas or liquid chromatography (GC or LC, respectively), followed by ionization and resolution according to the mass-to-charge ratio. MS methods have a high sensitivity and can detect secondary metabolites at low concentrations, making them more suitable for high-throughput methods [10,38]. The most used method for VOMs analysis is the headspace (HS) solid-phase microextraction (SPME) coupled with GC–MS, due to its reliable and reproducible results. SPME, developed by Arthur and Pawliszyn in the early 1990s [47,48], involves the partitioning of analytes from the sample solution into the sorbent coating of the SPME fiber due to the intermolecular interaction with the sorbent material [49]. This technique combines sampling, extraction, concentration, and sample preparation into a single step [47,50]. It is a highly efficient technique, with increased sensitivity, automation, and portability, that does not require any concentration step before analysis, thus preventing the production of interferents [51,52]. The availability of extraction materials and basic equilibration mechanisms makes this methodology very selective, fast, and cost-efficient, and it gives a high-level performance [47,48].

Different approaches have been proposed concerning volatilomic studies aiming to find a relationship between VOMs’ signature of the body and cancer, based on the comparison of the VOMs pattern in biological samples from cancer patients and healthy individuals. Many studies have focused on the identification of a cancer-characteristic odor fingerprint from biological fluids through the application of sensorial analyses via electronic noses (e-noses). E-noses consist of non-specific sensors that interact differently with VOMs. Each VOM generates a characteristic fingerprint due to the interaction with the sensor array that is then analyzed by an appropriate pattern-recognition system to investigate its nature and origin (reviewed by Reference [53]). Urine analysis via an e-nose, for instance, has been shown to distinguish between different types of cancer. This is a non-invasive method based on the finding that dogs can be trained to smell urine and, thus, recognize several types of cancers [54,55]. Other studies have suggested a chemical characterization of biological fluids for the identification of cancer-specific biomarkers through analytical techniques such as MS-based techniques or SPME [56]. These studies are often performed in readily accessible biofluids, mainly exhaled breath, urine, and saliva, since the sampling procedure of these matrices is non-invasive, painless, easy, does not cause any discomfort to the patients, and does not require any specific expertise or specialized staff [42,44]. Blood has also been used in VOMs studies, but obtaining blood samples is an invasive, costly, and time-consuming procedure. Moreover, changes in the temperature or pH of blood samples can change the VOMs profile [43]. Table 2 shows a few studies on the volatile composition of oncological patients for the identification of discriminative biomarkers for different cancer types in urine, exhaled breath, and saliva samples.

VOMs can originate from endogenous metabolomic pathways and/or from external sources, such as diet, drugs, and environmental exposure (exogenous). VOMs constitute a rich source of information on health or disease status, reflecting biochemical and metabolic activities, along with environmental effects, caused by biological activities, including cell death, oxidative stress, or inflammation [43].

VOMs in urine are considered intermediate or end products of metabolic pathways and may include ketones, alcohols, furans, and sulfides [44,72]. The human urinary profile changes over time due to bacterial activity, metabolism, pH variations, decomposition of urine constituents, health status, or physical stress. All of these factors are important sources of VOMs produced endogenously. In contrast, dietary habits and environmental exposure to contaminants are two important exogenous sources of many VOMs found in the human organism [51]. Urine is the preferred biological fluid for a volatilomic approach, due to the enrichment of volatile components, ranging in polarity and complexity [46], caused by their concentration in the kidney before excretion [51]. Moreover, its sampling can be performed as often as needed [59], it is easier to obtain in large quantities and to handle, it needs less sample preparation, and it contains high amounts of metabolites and low protein content [41].

Most studies on cancer-related VOMs have been performed by using exhaled breath, especially for the detection of lung cancer [62,63] (Table 2). Exhaled breath reflects the volatile composition of the bloodstream and airways and, thus, the status and condition of the whole metabolism. Hence, it has the potential to assess the diagnosis, severity, and progression of diseases [73]. Exhaled breath contains VOMs from endogenous sources, as well as a large number from exogenous origins. The endogenous metabolites are blood-borne compounds released to the environment through the lungs or compounds made from all classes of symbiotic bacteria. The exogenous VOMs include compounds inhaled from the external environment, including compounds produced after the oral ingestion of food and compounds derived from smoking cigarettes and exposure to pollutants and chemicals [44]. VOMs in exhaled breath can be easily detected through biological sensors, such as e-noses [44,74]; however, such an approach often targets a limited number of VOMs previously defined as having a discriminative potential for a given condition. A comprehensive analysis of the volatile composition of breath often requires laboratory layouts similar to those used for other volatilomics analyses. Proton transfer reaction with time-of-flight mass spectrometry (PTR-TOF–MS) allows for the real-time identification of the breath volatile composition, representing the most potent and sophisticated approach currently available [75]. Nevertheless, it involves high acquisition and maintenance costs, as well as specific adaptations to be used in the clinical environment and highly trained personnel to operate it (reviewed in Reference [73]). Given its informative potential to guide clinical decisions, continuous technological improvements will certainly drive the real-time monitoring of exhaled breath in the clinical environment, making it an important tool to add to conventional medical diagnostics [73].

Saliva is the easiest way of sampling biofluids to obtain relevant metabolic information. It is readily available in large amounts all day and contains fewer proteins than blood, thus decreasing any potential risk of non-specific interference and hydrostatic interactions [69,70]. Saliva is considered a mirror of the metabolic interaction with the environment, as it integrates both endogenous and exogenous contributions [76]. The volatile composition of saliva reflects the oral composition and the biochemical and metabolic blood information, constituting a valuable source of VOMs for cancer biomarkers [77,78,79]. This approach is still relatively new, and it is not as popular in medical diagnosis as urine and serum, but a few studies have demonstrated its potential for head and neck [67] and breast cancer [69,70] diagnosis.

#### Challenges of Volatilomics

Volatilomic studies comprise different approaches, such as the identification of a cancer-characteristic odor fingerprint from biological fluids through e-noses or the chemical characterization of biological fluids for the identification of cancer-specific biomarkers through MS-based techniques combined with multivariate statistical analysis [56]. These approaches are still in progress, and there are a few limitations that prevent them from being used in clinical practice.

E-noses typically use non-specific sensors and can only detect specific molecular patterns based on the differentiation of the odors’ fingerprints, which can vary substantially in different biological fluids. Consequently, it is possible to have different sets of VOMs in different biological fluids that are related to the same disease [54,55]. The discrimination between different odors is not operated based on the identification of their chemical composition, since e-noses cannot identify and quantify every VOM found in a sample; thus, other analytical techniques are also used, such as GC–MS. Hence, e-noses do not for allow the identification of specific biomarkers associated with a certain disease [53]. Further research is needed to develop and improve specific e-noses for different diseases, along with the need to discover and analyze the connections between specific diseases and body fluids’ odors.

MS-based techniques, such as GC–MS, also show some limitations that prevent them from being used in real-time diagnostic applications, such as low sample throughput, high costs, and the requirement for trained personnel and sophisticated software [80]. Moreover, these techniques are not portable, and differences in sample preparation and the lack of standardized analytical procedures and statistical treatment of data can compromise the comparison of results among different studies, which is a common problem and debated subject in the field of VOMs assessment [41,56,81]. In the future, the standardization of procedures from collection to data treatment might revolutionize the volatilomics field. For instance, given these concerns, Aggarwal et al. [46] proposed an optimized method for urine sample preparation followed by HS-SPME/GC–MS analysis.

The exhaustive comprehension of metabolic pathways and VOMs’ origin, in addition to a better evaluation of confounding factors’ influence, is another important point in the volatilomics approach. VOMs profile may vary according to the patient’s stage of cancer, and potential biomarker candidates must consist of endogenous VOMs, linked to disease-related changes in metabolism. Thus, it is crucial to select and separate endogenous VOMs from exogenous ones before the VOMs analyses. This selection avoids contamination from exogenous and uncontrolled sources, such as diet, medications, environmental factors, smoking, or alcohol consumption, which can lead to abnormal metabolism with subsequent excretion of differing concentrations of VOMs in the biofluids. Additionally, epigenetic factors play a very important role in the clinical phenotypes of cancer, meaning that the volatilomic biosignature and the possible biomarkers will differ between different regions of the world, due to genetic, environmental, and toxicological factors, in addition to the different eating habits practiced around the world, that can be related to the development of cancer. Gaining knowledge of the biochemical pathways involved in the VOMs formation is highly desirable to understand the formation of these metabolites and thus, determine their source since some of these metabolites can be originated from both endogenous and exogenous sources [44]. Therefore, all of these confounding factors make VOMs determination analytically challenging [56,81]. The progress in volatilomics studies will allow an exhaustive comprehension of metabolic pathways and an elucidation of the mechanisms of cancers and how they affect VOMs’ production. Moreover, this information will allow researchers to determine the VOMs’ origin in more detail, in addition to leading to a better evaluation of the confounding factors’ influence.

Despite these limitations, the tremendous informative potential of volatilomics will allow researchers to gain more in-depth knowledge of cancer development and progression. The volatilomics findings from GC–MS analyses will allow for the discovery of cancer-specific biomarkers and, consequently, the development of highly specific, fast, inexpensive, easy-to-use, and portable sensors [43]. Sensors do not require invasive procedures and can be easily implemented in clinical practice without the need for specialized staff [80]. Moreover, different types of sensors have been developed to detect cancer-related VOMs, including metal oxide and nanomaterial-based chemiresistive sensors, piezoelectric sensors, colorimetric sensors, and metal-organic frameworks, with very promising diagnostic accuracy in terms of specificity and sensitivity [43]. Given these advantages, this approach can be easily disseminated to countries where economic resources and advanced infrastructures are not available [10,41,42,44]. Therefore, despite the previously mentioned concerns, the standardization of methods, along with the development of highly specific sensors, will allow for the detection and quantification of specific metabolites for the definition of cancer biomarkers, thus proving the importance of the volatilomics approach [53,56].

### 3.2. Volatilome of Prostate Cancer—A Promising Approach for Biomarkers’ Detection

Prostate cells have a distinct metabolic profile, reflecting the production of citrate, PSA, and polyamines (spermine and myo-inositol), the major components of prostate fluid [33]. Studies on the metabolic alterations associated with PCa have demonstrated characteristic decreases in citrate and polyamines and increases in choline glycerophospholipids, lactate, and components of several pathways of amino acid metabolism [38], as well as in the synthesis and oxidation of fatty acids [10,82]. In the advanced stages of PCa, metastases formation is associated with an increase in the glycolytic pathway, also known as the Warburg effect. This phenomenon is characterized by a shift in energy production due to increased aerobic glycolysis and lactate secretion [10,83]. Dysregulations in 14 metabolic pathways mainly related to valine, leucine, and isoleucine biosynthesis; the pentose phosphate pathway; and glycine, serine, and threonine metabolism denoted that PCa development and progression are deeply connected to alterations in amino acids metabolism, energy metabolism, and membrane metabolism [10,84].

Cancer growth is promoted by the progressive accumulation of genetic, epigenetic, and post-translational changes and by a high metabolic demand, leading to cellular oxidative stress. In turn, these changes increase the liver’s production of Cytochrome P (CYP) 450 oxidase enzymes to deal with stress. CYP 450 enzymes catalyze the metabolism of many compounds of both exogenous and endogenous origin, including steroids, vitamins, fatty acids, prostaglandins, and leukotrienes. These enzymes may activate exogenous compounds to toxins or carcinogens, and mutations in the *CYP* genes can cause serious health problems. Given their ability to activate or deactivate most carcinogens, CYP 450 enzymes play an important role in cancer formation and are involved in tumor initiation or promotion [43]. For instance, mutations in *CYP17A1* lead to mineral corticoid excess syndromes, glucocorticoid, and sex-hormone deficiencies, increasing the risk of PCa and benign prostatic hypertrophy [85,86]. CYP 450 enzymes can also mediate the generation of oxygen reactive species, which are known to be overexpressed in cancer cells [43]. Oxidative stress also leads to lipid peroxidation, caused by disease processes or immune responses such as inflammation, and all of these changes can be detected through VOMs. VOMs are end products of carbohydrate and lipid metabolism, as well as oxidative stress and CYP 450 enzymes, and these processes can lead to different volatile profiles in cancer patients compared to cancer-free individuals (Figure 4).

The volatilomics is still relatively new in PCa when compared to other cancers (Table 2). Table 3 describes volatilomic studies for the definition of PCa biomarkers in urine, exhaled breath, and saliva. Most studies found were performed in urine, as this biofluid contains compounds coming directly from the prostate gland and does not require cross-over blood–tissue barriers, having fewer confounding elements [7]. The identification of specific biomarkers for cancer diagnosis consists of the chemical characterization of liquid urine or its headspace, through MS-based techniques or SPME, aiming at the detection of PCa biomarkers and quantification of their amounts [56]. Most research is based on the comparative analysis of samples from PCa patients and controls, such as the studies of Lima et al. [39], Khalid et al. [87], Stuck-Lewicka et al. [88], Gao et al. [89], and Jiménez-Pacheco et al. [90] (Table 3). These studies developed their methods by combining different analytical techniques, mostly HS-SPME/GC–MS, for the detection and quantification of changes in VOMs levels in PCa samples compared to healthy ones. Interestingly, the studies of Lima et al. [80] and Peng et al. [91] reported biomarkers for the differential diagnosis of PCa when compared with other cancers. Lima et al. [80] developed a urinary 10-biomarker panel for the diagnosis of PCa, with a higher accuracy level than the PSA test. This panel of biomarkers was able to discriminate PCa patients from controls and other urological cancers, including bladder and renal cancers. Peng et al. [91] focused on the volatile composition of PCa patients’ exhaled breath. In this study, the authors used a nanosensor array to discriminate the exhaled breath profile of lung, breast, prostate, and colorectal cancers. The sensor developed was able to differentiate between healthy controls and cancerous patients and between different cancer types. Nevertheless, no exhaustive results have been published until now, since many different VOMs have been proposed as PCa biomarkers, and divergent opinions upon the same metabolites have emerged in different studies.

Urine analysis via an e-nose has been shown to distinguish between PCa patients and healthy controls, according to their volatilome profiling. Filianoti et al. [54], Taverna et al. [92], Capelli et al. [93], and Bannaga et al. [94] used different methodologies based on urine analysis through e-noses (Table 3) and proved that urine headspace and its modification are connected to cancer. In turn, Waltman et al. [95] used an e-nose to distinguish between PCa patients and healthy controls, according to the volatilome profiling of exhaled breath. However, to our best knowledge, so far, no study has analyzed the VOMs found in the saliva of PCa patients.

Despite the many advantages of using a volatilomics approach to screen PCa, it is challenging to find robust biomarkers given the disparities in results between studies. Potential biomarker candidates must consist of endogenous VOMs linked to disease-related changes in metabolism. Additionally, epigenetic factors play a very important role in the clinical phenotypes of PCa, meaning that the volatilomic biosignature and the possible biomarkers will differ between different regions of the world, as previously explained [44]. Hopefully, the progressive increase in studies involving the VOMs composition of PCa patients will help to unveil biomarkers suitable for its diagnosis, as a complementary tool to the current methods [15]. Recent studies have shown that the conjugation of PSA screening with other methodologies, such as risk calculators, biomarkers, and imaging tests, e.g., magnetic resonance imaging (MRI) or fusion biopsies, can attenuate overdiagnosis and under-detection issues [96]. This means that the combination of volatilomics with other methodologies could be extremely valuable for the classification and screening of cancer, being beneficial in the active surveillance of patients [7]. In the future, the identification of the PCa volatilomic biosignature through the differentiation of clinical phenotypes in a group of patients, along with the use of specific sensors in the clinical practice, will allow for the stratification of individuals into subgroups on which outcomes and treatments are based, thus providing more personalized treatments [7].

## 4. Conclusions

The current PCa screening techniques have low accuracy in predicting the clinical behavior of tumors. This often leads to the overtreatment of indolent tumors, with important physical and psychological burdens, thus negatively affecting the patients’ quality of life and their adherence to further treatments. Studies show that the combination of PSA screening with other methodologies, such as risk calculators, imaging techniques, and biomarkers, can attenuate overdiagnosis and under-detection issues, possibly reducing the number of unnecessary biopsies. Recently, volatilomics has emerged as a promising tool for the definition of cancer biomarkers. This approach is based on the study of VOMs, which reflect the metabolic and biochemical changes related to disease progression. VOMs are a valuable source of information on overall health and are present in readily accessible biofluids such as saliva, urine, and exhaled breath. Empirical data have confirmed the potential of VOMs analysis for cancer screening, characterization of disease progression, and follow-up of the treatment’s success, as well as for the discrimination between different types of cancer. Many studies have focused on the identification of a cancer-characteristic odor fingerprint from biological fluids through the application of sensorial analyses (e-noses), while others suggest a chemical characterization of biological fluids for the identification of cancer-specific biomarkers. However, no exhaustive results have been published until now, since different VOMs have been proposed as PCa biomarkers, and divergent opinions upon the same metabolites emerged in different studies. Methods must be standardized from collection to data processing, and they should be performed with larger cohorts, preferably with patients from different countries and ethnicities, to better evaluate the influence of external factors, such as epigenetics, diet, medication, genetics, and environmental exposure. Moreover, the tremendous informative potential of volatilomics will allow for more in-depth knowledge of the biochemical pathways involved in cancer development and progression, in addition to an in-depth understanding of VOMs’ origin and their metabolic pathways [55]. In turn, these findings will allow for the establishment of cancer-specific biomarkers and will result in the development of highly specific, fast, inexpensive, easy-to-use, and portable sensors to be implemented in clinical practice. Hence, the volatilomic approach will be a valuable complementary tool to the current PCa screening methods.

## Figures and Tables

**Figure 1 cancers-14-03982-f001:**
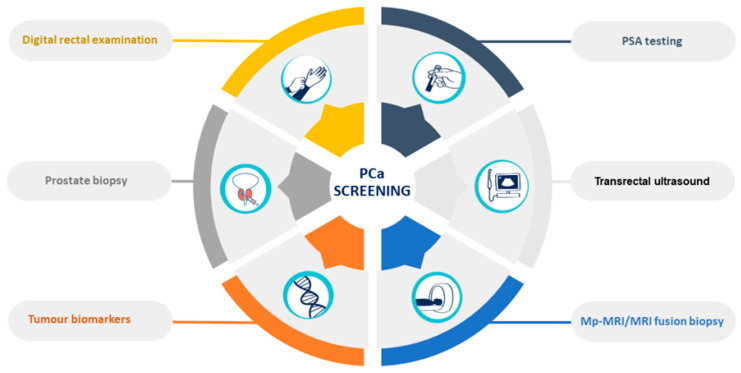
Prostate-cancer screening methods.

**Figure 2 cancers-14-03982-f002:**
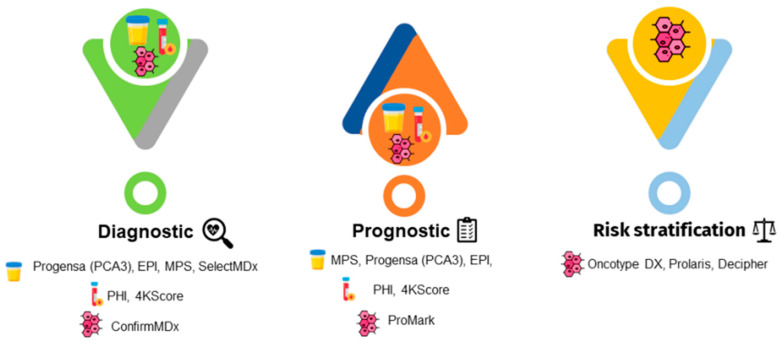
Emergent prostate cancer biomarkers for urine, serum, and tissue samples have been developed and can be used for diagnostic, prognostic, or risk stratification purposes. Legend: EPI, ExoDx Prostate IntelliScore; MPS, MyProstateScore; PCA3, prostate cancer gene 3; PHI, Prostate Health Index; 4KScore, Four-Kallikrein test.

**Figure 3 cancers-14-03982-f003:**
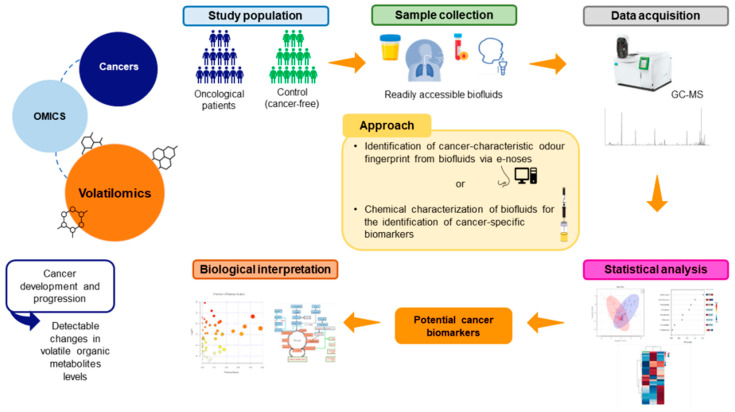
General flowchart of volatilomics approaches.

**Figure 4 cancers-14-03982-f004:**
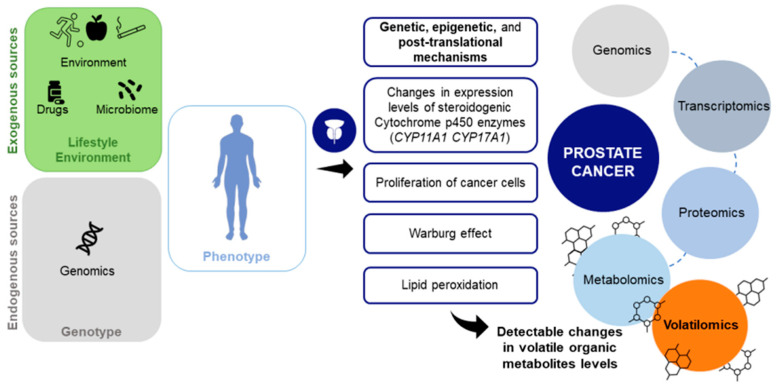
Cancer-related metabolic and biochemical activities can be detected through volatile organic metabolites. The establishment of a volatilomic biosignature can lead to the discovery of biomarkers for prostate cancer diagnosis.

**Table 1 cancers-14-03982-t001:** Potential clinical utility, characteristics, and availability of prostate cancer biomarkers.

Biomarker Test	Molecular Markers	Potential Clinical Utility	Characteristics	Availability
		Serum biomarkers		
PSA	PSA	Treatment monitoring	Sensitivity: 60% [18] Specificity: 79% [18] AUC: 0.55 [19]	
4KScore	Total PSA, free PSA, intact PSA, hK2	Unnecessary biopsy reduction of 43% [20] Risk prediction of PCa metastases Previous negative biopsy	Sensitivity: 75% [21] Specificity: 63% [21] AUC: 0.71 [22]	CLIA-certified
PHI	Total PSA, free PSA, p2PSA isoform	Unnecessary biopsy reduction of 40% [23] Prediction of high-grade PCa Active supervision monitoring	Sensitivity: 82% [24] Specificity: 80% [24] AUC: 0.71 [21]	FDA-approved
		Urinary biomarkers		
Progensa (PCA3)	Long non-coding RNAs (ratio of PCA3 mRNA:PSA mRNA)	Unnecessary biopsy reduction of 23–38% [25] PCa detection, staging, and prognosis Previous negative biopsy	Sensitivity: 69% [26] Specificity: 65% [26] AUC: 0.73 [26]	FDA-approved
SelectMDx	HOXC6 and DLX1 mRNA	Unnecessary biopsy reduction of 53% [27] Prediction of high-grade PCa	Sensitivity: 91% [27] Specificity: 36% [27] AUC: 0.71–0.81 [27]	CLIA-certified
MPS	PCA3 and TMPRSS2-ERG mRNA	Unnecessary biopsy reduction of 35–47% [28] Predict the risk of PCa and high-grade PCa	Sensitivity: 93% [29] Specificity: 33% [29] AUC: 0.69 [28]	CLIA-certified
EPI	Exosomal RNA (SPDEF, PCA3, ERG)	Unnecessary biopsy reduction of 27% [30] Improved identification of high-grade PCa	Sensitivity: 92% [30] Specificity: 34% [30] AUC: 0.70 [30]	CLIA-certified
		Tissue biomarkers		
ConfirmMDx	DNA hypermethylation GSTP1, APC, and RASSF1	Prediction of true negative prostate biopsies	Sensitivity: 68% [24] Specificity: 64% [24] AUC: 0.74 [24]	Not FDA-approved yet
OncotypeDX	mRNA expression (17 genes)	Monitoring of tumor aggressiveness	AUC: 0.73 [26]	Not FDA-approved yet
Prolaris	mRNA expression (31 genes)	Monitoring of tumor aggressiveness	AUC: 0.78 [26]	FDA-approved
Decipher	mRNA expression (22 genes)	Treatment monitoring	Sensitivity: 73% [31] Specificity: 74% [31] AUC: 0.79 [31]	CLIA-certified
ProMark	Protein biomarker test (8 proteins)	Monitoring of tumor aggressiveness	Sensitivity: 90% [27] Specificity: 85% [27] AUC: 0.72 [27]	CLIA-certified

Legend: AUC—area under the receiver operating characteristic (ROC) curve; CLIA—Clinical Laboratory Improvement Amendments; FDA—Food and Drug Administration.

**Table 2 cancers-14-03982-t002:** Studies on volatile organic compounds for the identification of cancer biomarkers in non-invasive matrices.

Cancer Type	Analytical Approach	Biomarker’s Candidates	Prediction Model	Validation Characteristics	Reference
Urine					
Head and neck	HS-SPME/GC–MS	*m*-cresol, benzene, nonanal, acetone	PLS-DA	NA	[57]
Head and neck	HS-SPME/GC–MS	2,6-dimethyl-7-octen-2-ol, 1-butanol, *p*-xylene, 4-methyl-2-heptanone	PLS-DA, ROC	NA	[58]
Leukaemia, colorectal, lymphoma	dHS-SPME/GC–qMS	16 VOMs were found statistically significant	PCA	NA	[59]
Breast	dHS-SPME/GC–qMS	Heptanal, dimethyl disulfide and 2-methyl-3-phenyl-2-propenal	PCA	NA	[51]
Renal cell carcinoma	HS-SPME/GC–MS	11 VOMs	PCA, PLSDA	NA	[60]
Pancreatic	TD-GC–TOF-MS GC–IMS	2,6-dimethyl-octane, nonanal, 4-ethyl-1,2-dimethyl-benzene, 2-pentanone	Repeated 10-Fold CV	NA	[61]
Exhaled breath					
Lung	HS-SPME/GC–MS	Acetone, methyl acetate, isoprene, methyl vinyl ketone, cyclohexane, 2-methylheptane, cyclohexanone	DFA, ANN	Sensitivity: 80% Specificity: 91.23% AUC: NA	[62]
Lung	HS-SPME/GC–MS	Caprolactam and propanoic acid	PCA, OPLS-DA, PLSDA	NA	[63]
Pancreatic	TD-GC–MS	Formaldehyde, acetone, acetoin, undecane, isopropyl alcohol, pentane, n-hexane, 1-butanol, 1-(methylthio)-propane, benzaldehyde, tetradecane, amylene hydrate	ROC	Sensitivity: 81% Specificity: 58% AUC: 0.736	[64]
Colorectal	TD-GC–MS	15 specific VOMs	PNN	Sensitivity: 86% Specificity: 83% AUC: 0.852	[65]
Gastric	PTR-TOF-MS	Propanal, aceticamide, isoprene, 1,3-propanediol	ROC	Sensitivity: 61% Specificity: 94% AUC: 0.842	[66]
Saliva					
Head and neck	HS-SPME/GC–MS	1,4-dichlorobenzene, 1,2-decanediol, 2,5-bis1,1-dimethylethylphenol, E-3-decen-2-ol	ROC, OPLS-DA	NA	[67]
Colorectal/stomach	GC–FID	Acetaldehyde, acetone, 2-propanol, ethanol, methanol	ROC	Sensitivity: 95.7% Specificity: 90.9% AUC: 0.857/0.839	[68]
Breast	HS-SPME/GC–MS	3-methyl-pentanoic acid, 4-methyl-pentanoic acid, phenol, p-tert-butyl-phenol (Portuguese samples) and acetic, propanoic, benzoic acids, 1,2-decanediol, 2-decanone, decanal (Indian samples)	PLS-DA, OPLS-DA	NA	[69]
Breast	dHS-SPME/GC–qMS	Phenol, 2-ethyl-1-hexanol	PCA	NA	[70]
Oral	HS-SPME/GC–MS	1-octen-3-ol, hexanoic acid, E-2-octenal, heptanoic acid, octanoic acid, E-2-nonenal, nonanoic acid, 2,4-decadienal, 9-undecenoic acid	PCA	Sensitivity: 100% Specificity: 100% AUC: 1	[71]

Legend: ANN, artificial neural network; AUC, area under the receiver operating characteristic (ROC) curve; CV, cross-validation; DFA, discriminant function analysis; dHS-SPME, dynamic headspace solid-phase microextraction; GC–IMS, gas chromatography–ion migration spectroscopy; GC–MS, gas chromatography-mass spectrometry; GC–qMS, gas chromatography–quadrupole mass spectrometry; HS-SPME, headspace solid-phase microextraction; NA, not analyzed; PCA, principal component analysis; OPLS-DA, orthogonal partial least-squares discriminant analysis; PLS-DA, partial least-squares discriminant analysis; PNN, probabilistic neural network; PTR-TOF-MS, proton-transfer-reaction time-of-flight mass spectrometry; ROC, receiver operating characteristic; TD-GC–MS, thermal desorption gas chromatography-mass spectrometry; TD-GC–TOF-MS, two-dimensional gas chromatography and time of flight mass spectrometer.

**Table 3 cancers-14-03982-t003:** Studies on volatile organic compounds for the identification of prostate cancer biomarkers.

Sample Groups	Analytical Approach	Biomarker’s Candidates	Prediction Model	Validation Characteristics	Reference
Urine					
PCa: 59 HC: 43	HS-SPME/GC–MS	2,6-dimethyl-7-octen-2-ol, pentanal, 3-octanone, 2-octanone	Repeated 10-Fold CV, Repeated Double CV	NA	[87]
PCa: 58 HC: 60	HS-SPME/GC–MS	hexanal, 2,5-dimethylbenzaldehyde, 4-methylhexan-3-one, dihydroedulan IA, methylglyoxal, 3-phenylpropionaldehyde	PLS-DA, ROC	Sensitivity: 89% Specificity: 83% AUC: 0.904	[39]
PCa: 20 BC: 20 RC: 20 HC: 20	HS-SPME/GC–MS	methylglyoxal, hexanal, 3-phenylpropionaldehyde, 4-methylhexan-3-one, 2,5-dimethylbenzaldehyde, dihydroedulan IA, ethylbenzene, heptan-2-one, heptan-3-one, 4-(2-methylpropoxy)butan-2-one, methyl benzoate, 3-methyl-benzaldehyde	PLS-DA	Sensitivity: 76% Specificity: 97% AUC: 0.90	[80]
PCa: 32 HC: 32	GC–MS	VOMs involved in amino acids, purine, glucose, urea, Krebs cycle biochemical pathways	PCA, PLS-DA	NA	[88]
PCa: 108 HC: 75	SBSE/TD-GC–MS	11 VOMs	ROC	Sensitivity: 87% Specificity: 77% AUC: 0.86	[89]
PCa: 29 BPH: 21	HS-SPME/GC–MS	furan, *p*-xylene	-	NA	[90]
BC: 15 PCa: 55 HC: 36	GC–TOF-MS and GC–IMS	35 VOMs	ROC, Repeated 10-Fold CV	GC–IMS method Sensitivity: 87% Specificity: 92% AUC: 0.95 GC–TOF-MS method Sensitivity: 78% Specificity: 88% AUC: 0.94	[55]
PCa: 88 HC: 86	Urine headspace conditioning, followed by e-nose analysis	The study tested the ability of urinary volatilome profiling to distinguish patients with PCa from HC	ROC	Sensitivity: 85.2% Specificity: 79.1% AUC: 0.82	[92]
PCa: 133 HC: 139	Urine headspace conditioning, followed by e-nose analysis (Cyranose C320)	The study tested the ability of urinary volatilome profiling to distinguish patients with PCa from HC	PCA, ROC	Sensitivity: 82.7% Specificity: 88.5% AUC: 0.90	[54]
PCa: 132 HC: 60	Urine headspace conditioning, followed by e-nose analysis (Cyranose C320)	The study tested the ability of urinary volatilome profiling to distinguish patients with PCa from HC	PCA	Sensitivity: 82% Specificity: 87% AUC: NA	[93]
HCC: 31 PCa: 62 BC: 29 HC: 18	SPME, followed by analysis with polymer tabs sensor	The study tested the ability of urinary volatilome profiling to distinguish patients with PCa from HC	PCA, ROC	PCa detection Sensitivity: 70% Specificity: NA AUC: 0.70	[94]
Exhaled breath					
LC: 30 CC: 26 BTC: 22 PCa: 18 HC: 22	HS-SPME/GC–MS	6 VOMs for LC, 6 VOMs for CC, 5 VOMs for BTC, 4 VOMs for PCa	PCA	NA	[91]
PCa: 32 HC: 53	E-nose analysis (Cyranose C320)	The study tested the ability of exhaled breath volatilome profiling to distinguish patients with PCa from HC	ANN	Sensitivity: 84% Specificity: 70 AUC: 0.79	[95]

Legend: ANN, artificial neural network; AUC, area under the ROC curve; BC, bladder cancer; BPH, benign prostate hyperplasia; BTC, breast cancer; CC, colon cancer; CV, cross-validation; GC–IMS, gas chromatography–ion mobility spectrometry; GC–MS, gas chromatography–mass spectrometry; GC–TOF-MS, gas chromatography coupled to time-of-flight mass spectrometry; HC, healthy control; HCC, hepatocellular cancer; HS-SPME, headspace solid-phase microextraction; kNN, *k*-nearest neighbor; LC, lung cancer; NA, not analyzed; PCA, principal component analysis; PLS-DA, partial least-squares discriminant analysis; RC, renal cancer; ROC, receiver operating characteristic; SBSE, stir bar sorptive extraction; TD-GC–MS, thermal desorption gas chromatography–mass spectrometry.

## Data Availability

Not applicable.

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
