# Peer review of "Volatilomics: An Emerging and Promising Avenue for the Detection of Potential Prostate Cancer Biomarkers"

_cancers, 2022, doi:10.3390/cancers14163982_

Round 1

Reviewer 1 Report

In the manuscript titled “Volatilomics: an emerging and promising avenue for the detection of potential prostate cancer biomarkers” the authors have reviewed several articles and suggested that volatile organic molecules can serve as a biomarker for prostate cancer.

The following major points need to be addressed before further consideration for publication. 

1.     I believe the authors have over-speculated the feasibility of the approach, volatilomics rely on mass spectrometry and chromatography techniques which are not as cheap and easy to use (need high expertise personnel to run the samples, operate the instruments and data analysis) as compared to other available techniques. So a routine clinical use of volatilomics could be challenging. I Just wanted to know the author's thought on it, if they have something different.  

2.     How volatilomics can help in the differential diagnosis of different cancer or among various subtypes of a cancer?  

3.     In clinics, what amount/ values of VOM’s can consider for healthy and diseased conditions? An example will be appreciated.

4.     How will a clinician make sure that the VOM’s generated in patients are due to disease conditions not because of their food habits or some environmental exposures.

5.     What are the limitations of volatilomics, and what are your suggestions to overcome them? I would like to add a section regarding that in the manuscript.

6.     It would be great if authors come up with a flowchart/ figures that can summarize the volatilomics for better understanding to readers. 

Author Response

Reviewer’s 1 comments:

« 1. I believe the authors have over-speculated the feasibility of the approach, volatilomics rely on mass spectrometry and chromatography techniques which are not as cheap and easy to use (need high expertise personnel to run the samples, operate the instruments and data analysis) as compared to other available techniques. So a routine clinical use of volatilomics could be challenging. I Just wanted to know the author's thought on it, if they have something different.»

«4. How will a clinician make sure that the VOM’s generated in patients are due to disease conditions not because of their food habits or some environmental exposures.»

«5. What are the limitations of volatilomics, and what are your suggestions to overcome them? I would like to add a section regarding that in the manuscript.»

Authors answer:

The authors thank the Reviewer’s comments. A new section was added to the manuscript discussing the limitations of volatilomics in more detail and suggestions to overcome them were also discussed. The following paragraphs provide the answers to the questions mentioned above.

Volatilomics is a subset of metabolomics based on the study of VOMs, which are generated through the metabolism of cells reflecting their biological activity. VOMs are released as metabolic products in readily accessible biofluids, such as exhaled breath, sweat, urine, or saliva [1]. In turn, volatilomics can provide comprehensive and detailed information on the phenotype and molecular physiology changes resultant from the interactions between environmental factors, genetics, and both exogenous and endogenous factors, such as age, diet, drugs, and chronobiological variations, among others [2].

Volatilomics studies comprise mainly two different approaches, aimed to find a relationship between VOMs’ signature of the body and cancer, that involve the comparison of the VOMs profile in biological samples from cancer patients and healthy individuals. The first one is based on the identification of a cancer-characteristic odour fingerprint from biological fluids through the application of sensorial analyses, via electronic noses (e-noses). The second approach consists of the chemical characterisation of biological fluids for the identification of cancer-specific biomarkers, through mass spectrometry (MS)-based techniques or SPME [3]. Both approaches are still in progress and there are a few limitations that prevent them to be used in clinical practice.

E-noses are based on sensors that interact differently with VOMs. Each VOM generates a characteristic fingerprint due to the interaction with the sensor array, which is analysed by an appropriate pattern recognition system to investigate its nature and origin [4]. Urine analysis via an e-nose, for instance, has been shown to distinguish between different types of cancer. This is a non-invasive method based on the findings dogs can be trained to smell urine and thus recognize several types of cancers [5,6]. However, e-noses typically use non-specific sensors and can only detect specific molecular patterns based on the differentiation of the odours’ fingerprints, which can vary substantially in different biological fluids. Consequently, it is possible to have different sets of VOMs in different biological fluids, related to the same disease [5,6]. The discrimination between different odours is not operated based on the identification of their chemical composition, since e-noses cannot identify and quantify every VOM found in a sample, for which other analytical techniques are used, like gas chromatography-mass spectrometry (GC-MS). Hence, e-noses do not allow the identification of specific biomarkers associated with a certain disease [4]. Despite showing very promising diagnostic accuracy in terms of specificity and sensitivity (table 3 of the manuscript), further research is needed to develop and improve specific e-noses for different diseases, along with the need to discover and analyse the connections between specific diseases and body fluids’ odours.

MS-based techniques, like GC-MS, have a high sensitivity and can detect secondary metabolites at low concentrations [7,8]. Although, these techniques also show some limitations that prevent them to be used in real-time diagnostic applications, such as low sample throughput and the requirement for trained personnel and sophisticated software [9]. Moreover, these techniques are expensive, not portable, and differences in sample preparation, the lack of standardized analytical procedures, and statistical treatment of data can compromise the comparison of results among different studies, which is a common problem and debated subject in the field of VOMs assessment [3,10,11]. In the future, standardization of procedures from collection to data treatment might revolutionize the volatilomics field. For instance, given these concerns, Aggarwal et al. [12] proposed an optimized method for urine sample preparation followed by HS-SPME/GC-MS analysis.

The exhaustive comprehension of metabolic pathways and VOMs’ origin, in addition to a better evaluation of confounding factors’ influence, is another important point in the volatilomics approach. VOMs profile may vary according to the patient’s stage of cancer and potential biomarker candidates must consist of endogenous VOMs, linked to disease-related changes in metabolism. Thus, it is crucial to select and separate endogenous VOMs from exogenous ones prior to the VOMs analyses. This selection avoids contamination from exogenous and uncontrolled sources, such as diet, medications, environmental factors, smoking, or alcohol consumption, which can lead to abnormal metabolism with subsequent excretion of differing concentrations of VOMs in the biofluids. Additionally, epigenetic factors play a very important role in the clinical phenotypes of cancer, meaning that the volatilomic biosignature and the possible biomarkers will differ between different regions of the world, due to genetic, environmental, and toxicological factors, in addition to the different eating habits practised around the world, that can be related to the development of cancer. The knowledge of the biochemical pathways involved in the VOMs formation is highly desirable to understand the formation of these metabolites, and thus, determine their source since some of these metabolites can be originated from both endogenous and exogenous sources [13]. Therefore, all these confounding factors make VOMs determination analytically challenging [3,11]. The progress in volatilomics studies will allow an exhaustive comprehension of metabolic pathways and an elucidation of the mechanisms of cancers and how they affect VOMs production. Moreover, this information will allow the determination of the VOMs’ origin in more detail, in addition to a better evaluation of confounding factors’ influence.

Despite these limitations, the tremendous informative potential of volatilomics will allow the knowledge in more depth of cancer development and progression. The volatilomics findings by GC-MS analysis will allow the discovery of cancer-specific biomarkers based on endogenous VOMs and, consequently, the development of highly specific, fast, inexpensive, easy-to-use, and portable sensors [1]. Sensors do not require invasive procedures and can be easily implemented in clinical practice without the need for specialized staff [9]. Moreover, different types of sensors have been developed to detect cancer-related VOMs, including metal oxide and nanomaterial-based chemiresistive sensors, piezoelectric sensors, colourimetric sensors, and metal-organic frameworks [1]. Given these advantages, this approach can be easily disseminated to countries where economic resources and advanced infrastructures are not available [8,10,14,15]. Therefore, despite the previously mentioned concerns, the standardization of methods along with the development of highly specific sensors will allow the detection and quantification of specific metabolites for the definition of cancer biomarkers, proving the importance of the volatilomics approach [3,4].

«2. How volatilomics can help in the differential diagnosis of different cancer or among various subtypes of a cancer?»

Cancer cells can be distinguished from normal cells by alterations in normal metabolic rates, apoptotic pathways, and protein expression patterns [1,8]. During cancer development and progression, the metabolomic shifts and the different responses of the immune system, lead to detectable changes in VOMs levels and might also result in the production of new VOMs, changing the volatile profile of an individual. Consequently, all these alterations in VOMs can be used to define a volatilomic biosignature for a specific cancer type [8,16].

Different cancer types produce distinct biomarker profiles, and endogenous VOMs have the potential to distinguish between them, as described in the studies in tables 2 and 3 of the manuscript and lines 465-473 of the revised manuscript. Some VOMs have been proposed to take part in some pathways involved in various types of cancer (general cancer biomarkers), while others only contribute to a particular type of cancer (cancer-specific biomarkers) (as reviewed by [1]). Moreover, urine analysis via an e-nose has been shown to distinguish between different types of cancer. E-noses detected specific molecular patterns based on the differentiation of the odours’ fingerprints [4]. These findings have shown that volatilomics can help in the differential diagnosis of different cancer types. Nevertheless, discriminating among various subtypes of a cancer can be more challenging to achieve.

«3. In clinics, what amount/ values of VOM’s can consider for healthy and diseased conditions? An example will be appreciated.»

Authors answer:

It is, undoubtably, an interesting question. As previously mentioned, some VOMs have been proposed to take part in some metabolomic pathways involved in various types of cancer (general cancer biomarkers), while others only contribute to a particular type of cancer (cancer-specific biomarkers) (as reviewed by [1]). These VOMs show a decreased or increased trend in oncological patients, depending on cancer type, when compared to the control group. However, in some cases different studies show nonconsensual results.

The volatilomic pattern give us information which allows to discriminate a specific type of cancer from healthy subjects. The building models based on multivariate statistical analysis allows you to correctly classify unknown cases based on the volatomic pattern. Unfortunately to date there are no absolute values for potential volatile biomarkers. Many more standardized studies, integrating many cases from different countries, will be necessary to determine the cut-off of potential volatile compounds biomarkers. It is not an easy task, as the epigenetic factors, which can influence the absolute values of volatiles, are many and very varied. So the challenge is huge.

  1. It would be great if authors come up with a flowchart/ figures that can summarize the volatilomics for better understanding to readers.

Authors answer:

A new figure (figure 3 in the revised version) was added to the manuscript that summarizes the volatilomics.

Reviewer 2 Report

The study is well done, the material is large enough and the methods look reliable. However the study is based on extensive and very recent literature, gives some new information and this warrants its publication.

Author Response

Reviewer’s 2 comments: 

«The study is well done, the material is large enough and the methods look reliable. However the study is based on extensive and very recent literature, gives some new information and this warrants its publication.»

Authors answer:

The authors thank the reviewer’ comments.

Round 2

Reviewer 1 Report

The authors have sufficiently addressed my concerns.